# Flexible parameter-sparse global temperature time-profiles that stabilise at 1.5 °C and 2.0 °C

Chris Huntingford[1], Hui Yang[2,1], Anna Harper[3], Peter M. Cox[3], Nic Gedney[4], Eleanor J. Burke[5], Jason A. Lowe[5], Garry Hayman[1], William J. Collins[6], Stephen M. Smith[7], and Ed Comyn-Platt[1]

[1]Centre for Ecology and Hydrology, Benson Lane, Wallingford, Oxfordshire, OX10 8BB, U.K.
[2]Department of Ecology, School of Urban and Environmental Sciences, Peking University, Beijing, P. R. China
[3]College of Engineering and Environmental Science, Laver Building, University of Exeter, North Park Road, Exeter, EX4 4QF, U.K.
[4]Met Office Hadley Centre, Joint Centre for Hydrometeorological Research, Maclean Building, Wallingford, OX10 8BB, U.K.
[5]Met Office, FitzRoy Road, Exeter, Devon, EX1 3PB, U.K.
[6]Department of Meteorology, University of Reading, Earley Gate, PO Box 243, Reading, RG6 6BB, U.K.
[7]Committee on Climate Change, 7 Holbein Place, London, SW1W 8NR, U.K.

*Correspondence to:* Chris Huntingford (chg@ceh.ac.uk)

**Abstract.** The meeting of the United Nations Framework Convention on Climate Change (UNFCCC) in December 2015 committed parties to the Convention to hold the rise in global average temperature to well below 2.0 °C above pre-industrial levels. It also committed the parties to pursue efforts to limit warming to 1.5 °C. This leads to two key questions. First, what extent of emission reductions will achieve either target? Second, what is the benefit of the reduced climate impacts by keeping warming at or below 1.5 °C? To provide answers, climate model simulations need to follow trajectories consistent with these global temperature limits. It is useful to operate models in an inverse mode to make model-specific estimates of greenhouse gas (GHG) concentration pathways consistent with the prescribed temperature profiles. Further inversion derives related emissions pathways for these concentrations. For this to happen, and to enable climate research centres to compare GHG concentrations and emissions estimates, common temperature trajectory scenarios are required. Here we define algebraic curves which asymptote to a stabilised limit, while also matching the magnitude and gradient of recent warming levels. The curves are deliberately parameter-sparse, needing prescription of just two parameters plus the final temperature. Yet despite this simplicity, they can allow for temperature overshoot and for generational changes where more effort to decelerate warming change is needed by future generations. The curves capture temperature profiles from the existing Representative Concentration Pathway (RCP2.6) scenario projections by a range of different earth system models (ESMs), which have warming amounts towards the lower levels of those that society is discussing.

## 1 Introduction

The conventional approach to understand climate change for possible different futures is to force earth system models with either emissions scenarios (e.g., Cox et al., 2000) or prescribed future atmospheric greenhouse gas concentrations (e.g., Mein-

shausen et al., 2011). However, recent UNFCCC meetings have placed a focus on prescribed temperature thresholds. This has mainly focused on how to avoid crossing 2.0 °C of global warming since pre-industrial times. Further, the December 2015 Paris Conference of the Parties (COP21) meeting suggested an additional aspiration of remaining below a 1.5 degrees warming threshold. To achieve the latter could in particular involve major changes in energy demand or production (Rogelj et al.,

2013), and extensive reliance on artificial carbon removal (Fuss et al., 2014) such as biofuels combined with carbon capture and storage. Equilibrium temperatures associated with even current GHG concentrations may already correspond to warming levels near to 1.5 °C (Huntingford and Mercado, 2016). Therefore given the likely difficulty of fulfilling the 1.5 °C target, there is a focus on understanding what is to be gained climatically from achieving that lower threshold, and the impacts of any temporary overshoot beforehand. There is a related need to calculate the amount of flexibility between different mixtures

of greenhouse gas emissions that will achieve the same eventual stabilisation levels. Forward modelling by prescription of emissions or GHG concentrations cannot answer these questions directly, as there is no guarantee that a particular simulation will asymptote precisely to an increase of 1.5 °C or 2.0 °C. Instead climate modelling needs to develop inversion methods that follow pre-defined future warming profiles. Existing ESM projections (e.g., from the CMIP5 database, Taylor et al., 2012) can be scaled to these, for instance by pattern scaling (e.g., Huntingford and Cox, 2000). Here we move towards that by presenting

families of temperature profiles that eventually stabilise. The use of common future warming trajectories may lead to easier discussion and comparison between projects designed to assess a range of implications of either the 1.5 °C or 2.0 °C target.

## 2   Temperature profiles that asymptote to prescribed temperature limits

### 2.1   One-parameter profiles

Derived are profiles of global warming above pre-industrial levels, $\Delta T(t)$ (°C), dependent on time $t$ (yr) and with $t = 0$ as

20  year 2015. Three boundary conditions are satisfied, with two related to present-day warming. One is an estimate of warming between pre-industrial times and the year 2015, $\Delta T_0$ (°C). The second is an estimate of the current rate of global warming, $\beta = \mathrm{d}\Delta T/\mathrm{d}t|_{t=0}$ (°C yr$^{-1}$). The values of these two parameters are derived from the HadCRUT4 dataset (Morice et al., 2012). We use the median from the 100 HadCRUT4 decadally-smoothed realisations of global temperature rise estimates (see Data Availability below; HadCRUT4 smoothing is with a 21 point binomial filter applied to annual values). Values in that

dataset normalise against the period 1961-1990; we renormalise to the period 1850-1900 as a proxy for pre-industrial times, giving $\Delta T_0 = 0.89$ °C. For further discussion of this value, see Hawkins et al. (2017). The recent gradient in warming is from regression fitting of the last 21 years (1995-2015 inclusive), giving $\beta = 0.0128$ °C yr$^{-1}$. We note, though, that when using HadCRUT4 as our observationally-based starting point, it is necessary to be aware of its non-global spatial extent. Additionally, it is compiled from a mix of air and sea surface temperatures, as described in Cowtan et al. (2015). The third

boundary condition is the final prescribed warming level $\Delta T_{\mathrm{Lim}}$ (°C), i.e. 1.5 °C or 2.0 °C. This is an eventual stabilisation level which our profiles $\Delta T$ approach asymptotically. The specification of the temperature thresholds in the COP21 statements could have other interpretations, including eventual stabilisation at even lower warming levels, or long-term temperature fluctuations

but which remain below prescribed limits. We do however allow the possibility of a near-term temporary overshoot of $\Delta T_{\text{Lim}}$, as described below.

We search for a parameter-sparse family of curves and consider a path that moves away from a linear temperature rise (via parameter $\gamma$) and towards a stabilisation level. Characterising different curves with an adaptation parameter $\mu$ (yr$^{-1}$) leads to:

$$\Delta T = \Delta T_0 + \gamma t - \left(1 - e^{-\mu t}\right)\left[\gamma t - (\Delta T_{\text{Lim}} - \Delta T_0)\right]. \tag{1}$$

A larger (positive) value for $\mu$ represents greater societal capability to adjust the temperature pathway towards a stable temperature state. The value of $1/\mu$ (yr) is an approximate e-folding time in moving from a non-zero positive gradient (in time) of global warming, and towards levelling off at $\Delta T_{\text{Lim}}$. Taking the time derivative of Eq. (1) (Appendix, Eq. (A2)) and matching to the historical record at year $t = 0$ gives:

$$\gamma = \beta - \mu\left(\Delta T_{\text{Lim}} - \Delta T_0\right). \tag{2}$$

Hence $\gamma$ is not the current rate of warming, i.e. $\gamma \neq \beta$. From Eq. (A4) and for $0 < \mu < 2\beta/(\Delta T_{\text{Lim}} - \Delta T_0)$, this gives $\mathrm{d}^2 \Delta T/\mathrm{d}t^2|_{t=0} < 0.0$, corresponding to no acceleration of the warming rate in the immediate future. Solutions require $\mu > 0$ for convergence.

Profiles for different $\mu$ and for $\Delta T_{\text{Lim}}$ values of 2.0 °C or 1.5 °C are presented in Fig. 1. For the three values selected, varying behaviours occur. The lower value of $\mu = 0.0074$ yr$^{-1}$ is sufficiently small that stabilisation can only be achieved after overshoot. The middle value of $\mu = 0.03$ yr$^{-1}$ achieves stabilisation without overshoot. The value of $\mu = 0.05$ yr$^{-1}$ also achieves stabilisation without overshoot; it corresponds to the strongest ability by society to adjust temperature. For this $\mu$ value, there is significant initial acceleration, particularly for $\Delta T_{\text{Lim}} = 2.0$ °C.

## 2.2 Two-parameter profiles

Whilst aiming to create profiles that are simple and mathematically tractable, allowing just one parameter may be overly restrictive. For example, society might be much more able to reduce emissions (corresponding to high $\mu$ values) in the further future, but may be less able in the near-term. To capture differences in generational approaches to fossil-fuel usage, one additional degree-of-freedom is introduced, setting $\mu(t)$ as a function of time:

$$\mu(t) = \mu_0 + \mu_1 t. \tag{3}$$

Matching the first derivative (Appendix, Eq. (A6)) at year $t = 0$ gives:

$$\gamma = \beta - \mu_0\left(\Delta T_{\text{Lim}} - \Delta T_0\right). \tag{4}$$

Profiles for different $\mu_0$ (yr$^{-1}$) and $\mu_1$ (yr$^{-2}$) are presented in Fig. 2. Curves can approach the warming target rapidly, then quickly asymptote to it through an increasingly large value in time of $\mu$ (e.g. red curve, 2.0 °C target). Similarly, increasing $\mu$ values offer the opportunity to have overshoot occurrences followed by rapid convergence to the desired warming level (e.g yellow curve, 1.5 °C target).

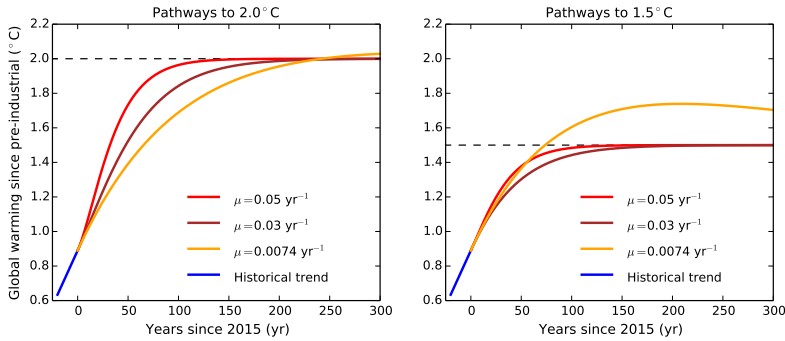

**Figure 1.** The effect of changing $\mu$ in the single-parameter temperature profiles, designed to asymptote to either 2.0°C (left panel) or 1.5°C (right panel). Values of $\mu$ as given in the legend.

The left-hand panel of Figure 3 presents the time from the year 2015 to achieve stabilisation, defined as within 0.01 °C of the target temperature threshold of 2.0 °C. The right-hand panel shows the maximum additional overshoot temperature, should $\Delta T_{\mathrm{Lim}}$ be crossed. Figure 4 shows the same for $\Delta T_{\mathrm{Lim}} = 1.5$ °C. These look-up charts enable the selection of a balance between general action on moving away from a business-as-usual approach to emissions (via parameter $\mu_0$) and leaving more change
to future generations (via parameter $\mu_1$). Lower $\mu_0$ and $\mu_1$ values take longer to reach stabilisation levels, although they risk temporary overshoot of the temperature target. The gray shading in the right-hand panels of Figs. 3 and 4 is where overshoot happens, and the temperature is rising throughout the 500-year period - hence peak warming occurs after that time. Overshoot is considered present if any year has a temperature above 0.01 °C of the target level. By definition, solutions of $\mu_0 < 0.0$ and $\mu_1 = 0$ never converge.
One potential evolution of global temperature could be a rapid rise to 2.0 °C of global warming, followed by strong efforts to reduce quickly to stabilisation at 1.5 °C. To achieve this on a single century timescale, with the curve structure of Eqs. (1), (3) and $\Delta T_{\mathrm{Lim}} = 1.5$°C, requires $\mu_0$ to be slightly negative, combined with high values of $\mu_1$. This influences the selection of the ranges of $\mu_0$ and $\mu_1$ in Fig. 4.

### 2.3 Fitting to existing ESM simulations

Equations (1), (3) and (4) generate a range of future temperature pathways towards prescribed warming limits. For these, the related changes in atmospheric gas concentrations and emissions can be determined. However many ESMs have been operated in forward mode, forced with scenarios of atmospheric greenhouse gas concentrations that correspond to heavy mitigation of fossil-fuel burning. The RCP2.6 scenario (Meinshausen et al., 2011) gives ESM-based estimates of the stabilisation of global warming around 2.0°C warming since pre-industrial times. We fit our model to these ESM projections of the RCP2.6 scenario.
Parameters $\beta$ and $\Delta T_0$ are tuned to their projections of temperature for the years 1995 to 2015 inclusive, whilst $\Delta T_{\mathrm{Lim}}$, $\mu_0$ and $\mu_1$ are fitted to the years 2016 to 2100. Figure 5 shows this curve calibration against three representative ESM RCP2.6 projections, expanded to the full set of 25 ESMs in Fig. A1. Across the years from 2016 to 2100 and for each individual ESM,

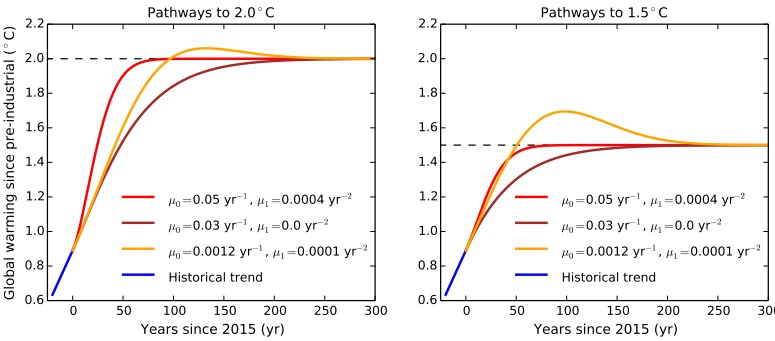

**Figure 2.** The effect of changing $\mu_0$ and $\mu_1$ in the two-parameter temperature profiles, designed to asymptote to either 2.0°C (left panel) or 1.5°C (right panel). Values of $\mu_0$ and $\mu_1$ as given in the legend.

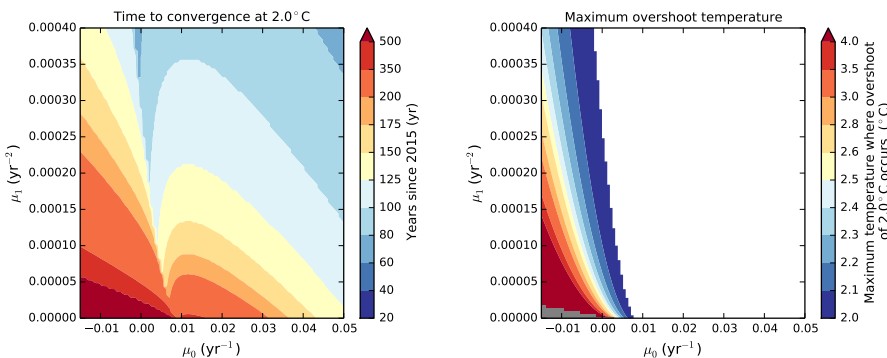

**Figure 3.** The dependence of the time to stabilisation and any overshoot magnitude (where present, white space otherwise) on the parameters $\mu_0$ and $\mu_1$ in the temperature profiles, and with $\Delta T_{\mathrm{Lim}}$=2.0°C. The scale of the colourbar is nonlinear. The gray region at the bottom left side of the right-hand panel is where temperatures become higher than the target of 2.0°C and are increasing throughout the 500 years, and so peak warming is not attained in that time.

the root mean square error (RMSE) of the differences between the fit and the ESM simulation is calculated. The mean of these RMSE values is 0.11 °C. This value is similar to the RMSE of differences between measurement and model estimates of global temperature interannual variability after detrending (e.g. Table 1b of Braganza et al., 2003). This confirms our curves can reproduce the RCP2.6 high mitigation ESM projections. Otherwise, any systematic differences would cause the RMSE
5    deviations to be higher than those of the interannual variability only; the latter is not represented in our profiles.

We additionally fit our curves to pathways in which the emissions are generated using integrated assessment models (IAM) and the related global temperature profiles are created using a simple climate model. This has been done for warming profiles from the IPCC scenario database (https://tntcat.iiasa.ac.at/AR5DB/) and for the marker scenarios of the more recent shared

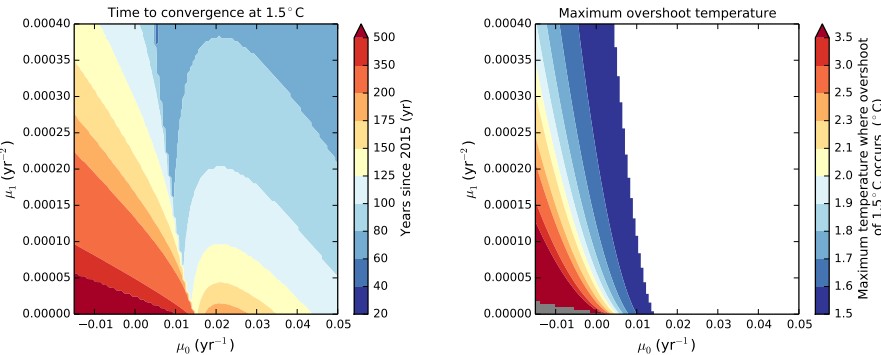

**Figure 4.** The dependence of the time to stabilisation and any overshoot magnitude (where present, white space otherwise) on the parameters $\mu_0$ and $\mu_1$ in the temperature profiles, and with $\Delta T_{\mathrm{Lim}}$=1.5°C. The scale of the colourbar is nonlinear. The gray region at the bottom left side of the right-hand panel is where temperatures become higher than the target of 1.5°C and are increasing throughout the 500 years, and so peak warming is not attained in that time.

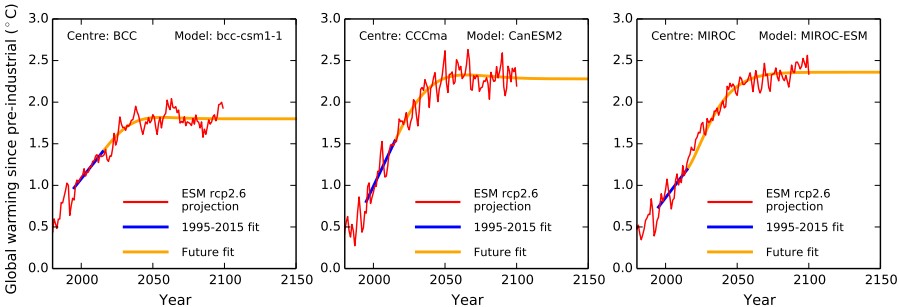

**Figure 5.** Fit of Eq. (1) (oranges curves) for the years after 2015, and for three representative ESM simulations (red curves) that correspond to the RCP2.6 scenario of atmospheric gas changes. The blue curve is the linear fit to the ESM for period 1995-2015. Annotated in each panel is the modelling centre and the ESM name. The fit to all the RCP2.6 simulations is given in Fig. A1.

socioeconomic pathways (SSP) database (https://tntcat.iiasa.ac.at/SspDb). We demonstrate that the functional forms used here can also represent these IAM-based scenarios to a good level of accuracy (see Supplementary Information).

## 2.4 Accounting for uncertainty in warming rates

The relatively low rate of warming increase since the year 1998 has been the subject of debate, and is sometimes referred to
as the "warming hiatus". The possibility of this occurring has been assessed in detail (e.g., Roberts et al., 2015). If a natural decadal-timescale fluctuation has temporarily suppressed the background warming trend, then our HadCRUT-based warming rate $\beta$ could be too small. The MAGICC climate impacts model, parameterised against a range of ESMs, typically projects the

recent warming as around $\beta = 0.025$ °C yr$^{-1}$. As a sensitivity study, we reproduce Fig. 3 and Fig. 4 using that higher warming rate, as Fig. A2 and Fig. A3, respectively.

## 2.5 Applications

Our profiles enable a common framework for the discussion of warming trajectories that stabilise to pre-defined temperature

limits. Regional climate change corresponding to these global temperatures can be estimated from interpolation of ESM projections (e.g. by pattern-scaling, Huntingford and Cox, 2000). Such scaling techniques can be linked to impacts models (e.g., Huntingford et al., 2010). In the comprehensive review of methods to identify regional differences associated with alternative global warming targets, James et al. (2017) note pattern-scaling as a key technique. The accuracy of this interpolation system has been recently reviewed in detail by Tebaldi and Arblaster (2014) and with enhancements proposed by Herger et al. (2015).

In the other approaches of James et al. (2017), the central issue remains as how to interpret existing simulations, that even for identical forcings, project a range of different future final warming levels.

Emissions profiles can be calculated to fulfil the ESM-dependent radiative forcings associated with any prescribed global temperature stabilisation profile. These can include different mixtures of individual greenhouse gas emissions, whilst accounting for any perturbed land-atmosphere and ocean-atmosphere gas exchanges. The sum of the radiation changes for altered

individual atmospheric greenhouse gas combinations must equal the ESM-dependent radiative forcing. Although our analytical forms are generic and can be calculated for any prescribed final stabilised temperature $\Delta T_{\mathrm{Lim}}$, the emphasis here is placed on the 1.5 °C or 2.0 °C targets. This is due to their strong current discussion in policy circles regarding "clean energy" (e.g. Obama, 2017).

To understand the significance between stabilizing global warming at either 1.5°C or 2.0°C is a complex and multi-

dimensional problem. There are implications for regional climate changes, impacts and for "allowable" emissions including the range of potential mixes between emitted greenhouse gases. These factors will also depend on the time evolution of global warming towards such warming thresholds. Each of these issues requires study, and ideally in a way that enables findings to be compared in a common framework. The application of these curves is to work towards such a framework, by offering a set of possible future warming pathways for utility in research initiatives, and that can be readily defined through a limited set of

parameters.

## 3 Conclusions

Presented in this work are parameter-sparse algebraic curves that match contemporary levels and the rate of change of global mean temperature, and which asymptote to prescribed warming thresholds. These represent a smooth transition from current rates of warming through to stabilised temperature levels. They can include an initial overshoot of temperatures above any

desired final warming level. Their relative simplicity makes them transparent, and open to discussion. If common temperature scenarios are adopted by a range of studies (by selection of $\mu_0$, $\mu_1$ and $\Delta T_{\mathrm{Lim}}$ values), this may allow easier comparison of

either the impacts of, or emissions to achieve, 1.5 °C or 2.0 °C warming stabilisation. At this stage, we do not associate any particular parameter combinations (or ranges) with their feasibility of fulfilment by society.

The curves have five parameters, with three of these constrained by: the current warming level $\Delta T_0$, the current rate of warming change $\beta$ and the final stabilised state $\Delta T_{\mathrm{Lim}}$. The remaining two parameters $\mu_0$ and $\mu_1$, offering two degrees of free-dom, give flexibility to the pathway shape before asymptoting to the temperature $\Delta T_{\mathrm{Lim}}$. Our curves allow for the possibility of temporary overshoot. This enables characterisation of the illustrative scenarios proposed in (Fig. 4 of Schneider and Mastran-drea, 2005), and their metric of dangerous anthropogenic interference (DAI) defined as the integrated time and magnitude spent overshooting a safe upper limit. Where an impacts study is for a period ahead that is much less than the time to stabilisation, then these curves allow for the possbility of gradually rising or declining temperatures through any analysis period.

Some very specific pathways may require further versatility. For instance, defining a pathway asymptoting to 1.5 °C and allowing warming overshoot to 2.0 °C constrains one degree of freedom. If the difference between the speed of approaching 2.0°C is specified as either much quicker or much slower than the time from that peak to 1.5°C, then two more degrees of freedom are required giving three in total. To satisfy situations such as this, further curve forms could, for instance, include specification of $\mu$ as a quadratic function of time.

## 4   Code availability

The python scripts leading to any of the diagrams is available on request to Chris Huntingford (chg@ceh.ac.uk)

## 5   Data availability

The global warming amount to the present day, along with the estimates of its gradient, comes from the HadCRUT dataset. In particular, the global annual anomalies are used from the median of the 100 member ensemble. These values are column 2 (column 1 is date) of: http://www.metoffice.gov.uk/hadobs/hadcrut4/data/current/time_series/HadCRUT.4.5.0.0.annual_ns_avg_smooth.txt

## Appendix A:  First and Second derivatives

Here we present the first and second derivatives for the one- and two-parameter profiles.

### A1   One-parameter profiles

The first derivative of Eq. (1) satisfies:

$$\frac{\mathrm{d}\Delta T}{\mathrm{d}t} = \gamma - \left(1 - e^{-\mu t}\right)[\gamma] - \left[-e^{-\mu t} \times (-\mu)\right]\left[\gamma t - (\Delta T_{\mathrm{Lim}} - \Delta T_0)\right] \tag{A1}$$

which at $t = 0$ gives:

$$\frac{\mathrm{d}\Delta T}{\mathrm{d}t}\Big|_{t=0} = \beta = \gamma + \mu(\Delta T_{\mathrm{Lim}} - \Delta T_0). \tag{A2}$$

The second derivative of Eq. (1) is found by differentiating Eq. (A1) with respect to $t$, giving:

$$
\begin{aligned}
\frac{d^2 \Delta T}{dt^2} &= -(-e^{-\mu t} \times -\mu)[\gamma] - \left[-e^{-\mu t} \times -\mu\right][\gamma] - \left[-e^{-\mu t} \times (-\mu) \times (-\mu)\right] \times [\gamma t - (\Delta T_{\text{Lim}} - \Delta T_0)] \\
&= -2\mu\gamma e^{-\mu t} + \mu^2 e^{-\mu t}[\gamma t - (\Delta T_{\text{Lim}} - \Delta T_0)]
\end{aligned}
$$

and which at time $t = 0$ gives:

$$
\frac{d^2 \Delta T}{dt^2}\Big|_{t=0} = -2\mu\gamma - \mu^2(\Delta T_{\text{Lim}} - \Delta T_0). \tag{A3}
$$

Substitution of condition (2) in to Eq. (A3) gives:

$$
\frac{d^2 \Delta T}{dt^2}\Big|_{t=0} = -2\mu\beta + \mu^2(\Delta T_{\text{Lim}} - \Delta T_0). \tag{A4}
$$

## A2  Two-parameter profiles

The first derivative of Eq. (1) with time-dependent $\mu$ as given in Eq. (3) satisfies:

$$
\frac{d\Delta T}{dt} = \gamma - \left(1 - e^{-[\mu_0 + \mu_1 t]t}\right)[\gamma] - \left[-e^{-[\mu_0 + \mu_1 t]t} \times (-\mu_0 - 2\mu_1 t)\right][\gamma t - (\Delta T_{\text{Lim}} - \Delta T_0)] \tag{A5}
$$

and which at $t = 0$ gives:

$$
\frac{d\Delta T}{dt}\Big|_{t=0} = \beta = \gamma + \mu_0(\Delta T_{\text{Lim}} - \Delta T_0). \tag{A6}
$$

The second derivative is found by differentiating Eq. (A5) with respect to $t$, giving:

$$
\begin{aligned}
\frac{d^2 \Delta T}{dt^2} &= -(-e^{-[\mu_0 + \mu_1 t]t} \times [-\mu_0 - 2\mu_1 t])[\gamma] - \left[-e^{-[\mu_0 + \mu_1 t]t} \times [-\mu_0 - 2\mu_1 t]\right][\gamma] \\
&\quad - \left[-e^{-[\mu_0 + \mu_1 t]t} \times (-\mu_0 - 2\mu_1 t) \times (-\mu_0 - 2\mu_1 t) - e^{-[\mu_0 + \mu_1 t]t} \times -2\mu_1\right] \times [\gamma t - (\Delta T_{\text{Lim}} - \Delta T_0)] \\
&= \left(-2[\mu_0 + 2\mu_1 t]\gamma + [(-\mu_0 - 2\mu_1 t)^2 - 2\mu_1][\gamma t - (\Delta T_{\text{Lim}} - \Delta T_0)]\right) \times e^{-[\mu_0 + \mu_1 t]t}.
\end{aligned}
$$

At time $t = 0$, this gives:

$$
\frac{d^2 \Delta T}{dt^2}\Big|_{t=0} = -2\mu_0\gamma - [\mu_0^2 - 2\mu_1](\Delta T_{\text{Lim}} - \Delta T_0). \tag{A7}
$$

## Appendix B:  Additional figures

Figure A1 repeats Fig. 5, but showing the fit of curves and related parameters ($\Delta T_{\text{Lim}}$, $\mu_0$, $\mu_1$, $\beta$ and $\Delta T_0$) for 25 ESM simulations of the RCP2.6 scenario. For these future fits, there is some interplay between parameter values that can achieve a good fit. The values fitted were constrained such that in all cases, $0.0 \leq \Delta T_{\text{Lim}} \leq 4.0$ °C, $-0.02 \leq \mu_0 \leq 0.08$ yr$^{-1}$ and $0.0 \leq \mu_1 \leq 0.0006$ yr$^{-2}$. A visual scan suggests a generally good fit for all ESMs except the GFDL_CM3 model.

Figure A2 shows the dependence of time to convergence and any overshoot amount on $\mu_0$ and $\mu_1$, whilst converging to 2.0 °C of global warming. The recent rate of warming is set to $\beta = 0.025$ °C yr$^{-1}$. Figure A3 is also for this higher $\beta$ value, and converging to 1.5 °C of global warming.

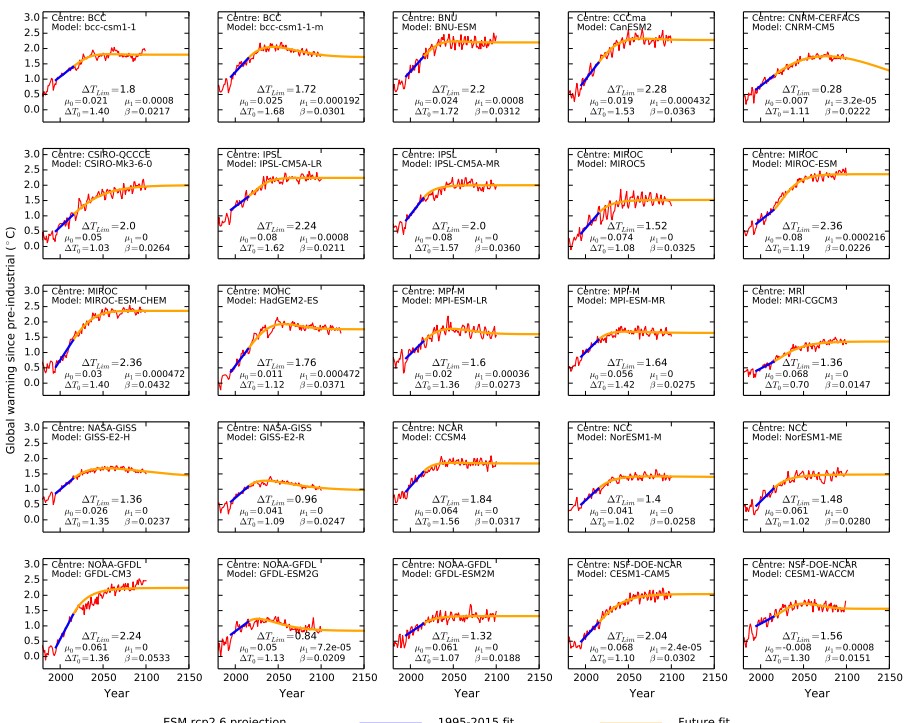

**Figure A1.** Identical to Fig. 5 except showing fitted curves for a larger set of 25 ESMs. Annotated in each panel is the modelling centre, ESM name and values of $\Delta T_{\mathrm{Lim}}$ ($^{\circ}$C), $\mu_0$ (yr$^{-1}$), $\mu_1$ (yr$^{-2}$), $\Delta T_0$ ($^{\circ}$C) and $\beta$ ($^{\circ}$C yr$^{-1}$)

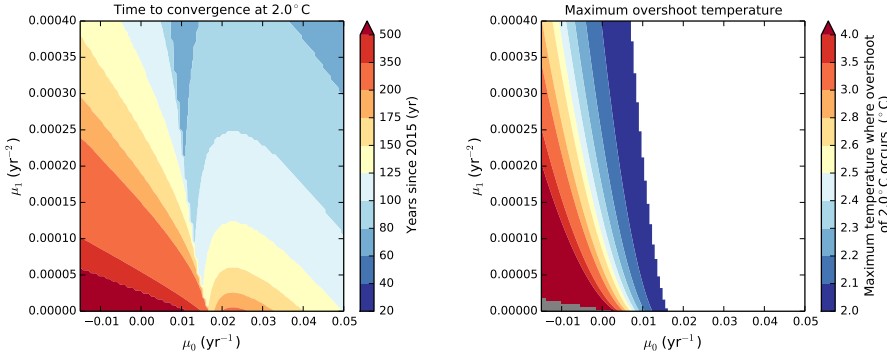

**Figure A2.** Identical to Fig. 3 but with $\beta = 0.025$ $^{\circ}$C yr$^{-1}$.

*Author contributions.* CH created the mathematical profiles and designed the paper. All authors helped discuss the expected requirements of the curves for research in to differences between achieving the 1.5 $^{\circ}$C and 2.0 $^{\circ}$C target. All authors made suggestions as to diagram format and aided in writing the paper.

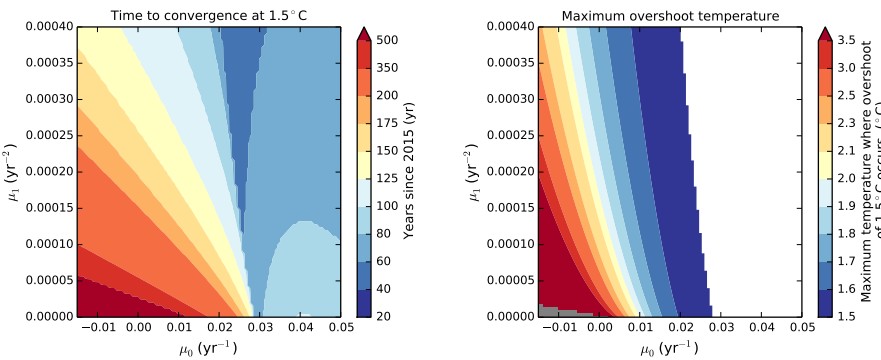

**Figure A3.** Identical to Fig. 4 but with $\beta = 0.025\ ^{\circ}\text{C yr}^{-1}$.

*Competing interests.* The authors confirm they have no competing interests.

*Acknowledgements.* C.H. acknowledges the NERC National Capability Fund. All authors (except SMS) received support from the U.K. Natural Environmental Research Council program "Understanding the Pathways to and Impacts of a 1.5 °C Rise in Global Temperature", through specific projects NE/P014909/1, NE/P014941/1 and NE/P015050/1. We acknowledge the World Climate Research Programme's Working Group on Coupled Modelling, which is responsible for CMIP, and we thank the climate modelling groups for producing and making available their model output.

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
