# Peer review of "Flexible parameter-sparse global temperature time-profiles that stabilise at 1.5 $^{\circ}$ C and 2.0 $^{\circ}$ C"

_Earth System Dynamics, 2017_

## Referee Comment (RC1) · Anonymous Referee #1 · 30 Mar 2017

General Comments:

Huntingford et al. develops simple temperature-time profiles that stabilize at 1.5 and 2.0 degrees C above preindustrial levels. In climate model comparisons of global warming, typically the greenhouse gas concentrations or emissions are held constant across models and the range of temperature responses is examined. This paper provides a first step towards a method to standardize global mean temperature progressions which could then be used to assess the range of greenhouse gas concentration or emission functions that are consistent with such global mean temperature progressions. The paper is modest in its goals but is worthy of publication.

Specific Comments:

[Figure]

Line 21 – Comparisons between HadCRUT4 and ESM-simulated global mean surface air temperature should probably take into account the non-global spatial extent of Had-CRUT4 as well as HadCRUT4's blend of air and sea surface temperatures (Cowtan et al., 2015)

-Cowtan, K.,ÂăZ. Hausfather,ÂăE. Hawkins,ÂăP. Jacobs,ÂăM. E. Mann,ÂăS. K. Miller,ÂăB. A. Steinman,ÂăM. B. Stolpe, andÂăR. G. WayÂă(2015),ÂăRobust comparison of climate models with observations using blended land air and ocean sea surface temperatures,ÂăGeophys. Res. Lett.,Âă42,Âă6526–6534, doi:10.1002/2015GL064888.

Line 22 – What does decadally smoothed mean? A 10-year running mean or some other filter?

Figure 1 and 2 – I would suggest limiting the x-axis maximum to the year 300 so that the differences in the curves over the 21st century can be seen more easily.

Technical Corrections:

Line 14 – I suggest that the authors change "but may be less able soon" to "but may be less able to in the near-term"

Figure 4 caption – Becom -> become

---

## Referee Comment (RC2) · Anonymous Referee #2 · 23 Apr 2017

General comments:

Huntingford and colleagues present a simple and transparent parametrization of temperature profiles that stabilize global mean temperature rise to a predefined level. They show how variations of two parameters can result in a wide variety of temperature profiles, with varying lengths of temperature overshoot. The authors suggest that these profiles can be used to better compare impact studies and that these profiles can be used to drive pattern scaling approaches. While I see no flaws in the mathematical description presented by the authors, there are several statements which have a weak factual basis, or for which evidence is missing. These statements require further analysis by the authors to show that their parametrization can capture temperature profiles

in both a useful and appropriate way.

Specific comments:

1) The validation of the appropriateness of the parametrization of temperature profiles is insufficient for the area of applicability. The authors claim that their profiles "enable a common framework for discussion of warming profiles that stabilize to pre-defined temperature limits", but provide no evidence other than being able to reasonably well fit to RCP2.6 simulations. To show its appropriateness as a common framework, the parametrizations should not only capture the response of multiple ESMs to one concentration profile, but also capture the multitude of concentration profiles available in the literature. The authors can deal with this by showing that their parametrization can be fitted to all temperature profiles of scenarios available in the IPCC scenario database (https://tntcat.iiasa.ac.at/AR5DB/), and the more recent SSP database (https://tntcat.iiasa.ac.at/SspDb/).

2) A smaller point is the dependence of the framework on stabilizing temperatures. For impact studies, it would also be interesting to be able to explore pathways which gradually decline temperatures.

3) P1L3: The Paris Agreement is committed to holding the rise in global average temperature increase to "well below" 2°C

4) P1L4-5: The "given emissions cuts to achieve the lower target may be especially difficult to achieve" argument is weak, and is not supported in the remainder of the manuscript. Depending on timing, similar emissions cuts are to be considered for 1.5°C and 2°C scenarios (but with a delay of 1 decade or so). In addition to the questions highlighted here, there is at least one more very important question, which is related to the reversibility of warming after an overshoot. I think this should also be mentioned.

5) P1L7: The basis for this "implication" is weak. Until now, models have been run in forward mode and have been able to provide lots of useful information for limiting

warming to and impacts at specific temperature limits.

6) P1L14: Please specify that these are projections by different ESMs.

7) P2L3-4: It would be good to provide a reference for this claim.

8) P2L4-5: Not clear what the relevance is of this statement. The authors refer here to the purely academic case of constant concentrations. Such a case is arguably in practice even harder to achieve than eliminating emissions.

9) P2L9-10: Inverse modelling also cannot answer these questions, because there is no way to ensure that pathways are supported by technologies.

10) P2L14-15: Recent publications provide an overview of various methods of exploring differences between warming levels of 1.5 and 2°C (James et al, Wiley Interdisciplinary Reviews: Climate Change, 2017). It would be useful to situate the approach proposed here in the context of these various methods.

11) P6L15ff: This section is unclearly written. Please consider rewriting it providing a bit background to how the suggested activities could be implemented.

12) P2&6: More recent papers have shown limitations of pattern scaling (e.g. Tebaldi and Arblaster, Clim. Ch., 2016). It would be good to also discuss these more recently identified limitations in the context of the proposed approach.

Technical corrections:

1) P7L8-9: This sentences seems incomplete. More open to scrutiny and discussion compared to what?

---

## Author Comment (AC1) · 19 May 2017

Huntingford et al. develops simple temperature-time profiles that stabilize at 1.5 and 2.0 degrees C above preindustrial levels. In climate model comparisons of global warming, typically the greenhouse gas concentrations or emissions are held constant across models and the range of temperature responses is examined. This paper provides a first step towards a method to standardize global mean temperature progressions which could then be used to assess the range of greenhouse gas concentration or emission functions that are consistent with such global mean temperature progressions. The paper is modest in its goals but is worthy of publication.

[Figure]

We thank this reviewer for their time and encouragement over this manuscript.

Line 21 – Comparisons between HadCRUT4 and ESM-simulated global mean surface air temperature should probably take into account the non-global spatial extent of Had-CRUT4 as well as HadCRUT4's blend of air and sea surface temperatures (Cowtan et al., 2015)

Cowtan et al (2015) "Comparison of climate models with observations using blended land air and ocean sea surface temperatures" Geophys. Res. Lett 42, 6526–6534, doi:10.1002/2015GL064888.

We use HadCRUT4 to provide an estimate of recent temperature rise $\Delta T0$ ($^\circ$C) and rate of temperature rise $\beta$ ($^\circ$C yr-1). This provides initial conditions to our mathematical curves. For precise policy applications, then we agree it is important to make the user aware of these issues. We now write in the manuscript: "We note, though, that when using HadCRUT4 as our observationally-based starting point, then it is necessary to be aware of its non-global spatial extent. Additionally it is compiled with a mix of air and sea surface temperatures, as described in Cowtan et al. (2015)."

Line 22 – What does decadally smoothed mean? A 10-year running mean or some other filter?

We have clarified this from the HadCRUT4 documentation, and now write in the manuscript: "HadCRUT4 smoothing is with a 21 point binomial filter applied to annual values"

Figure 1 and 2 – I would suggest limiting the x-axis maximum to the year 300 so that the differences in the curves over the 21st century can be seen more easily.

Done – the x-axis is now for the period up to 300 years from present day.

Technical Corrections: Line 14 – I suggest that the authors change "but may be less able soon" to "but may be less able to in the near-term"
Done

Figure 4 caption – Becom -> become

Corrected

---

## Author Comment (AC2) · 19 May 2017

General comments: Huntingford and colleagues present a simple and transparent parametrization of temperature profiles that stabilize global mean temperature rise to a predefined level. They show how variations of two parameters can result in a wide variety of temperature pro- files, with varying lengths of temperature overshoot. The authors suggest that these profiles can be used to better compare impact studies and that these profiles can be used to drive pattern scaling approaches. While I see no flaws in the mathematical description presented by the authors, there are several statements which have a weak factual basis, or for which evidence is missing. These statements require further analysis by the authors to show that their parametrization can capture

temperature profiles in both a useful and appropriate way.

We thank this reviewer, who has helped us to generate a better version of the paper. We have clarified with more rigour the statements highlighted by Referee #2. We have also performed the suggested further analysis by comparing our mathematical forms for temperature profiles against simulations in the IPCC scenario and SSP databases. Please see our full responses below, and including a new Supplementary Information.

Specific comments:

1) The validation of the appropriateness of the parametrization of temperature profiles is insufficient for the area of applicability. The authors claim that their profiles "enable a common framework for discussion of warming profiles that stabilize to pre-defined temperature limits", but provide no evidence other than being able to reasonably well fit to RCP2.6 simulations. To show its appropriateness as a common framework, the parametrizations should not only capture the response of multiple ESMs to one concentration profile, but also capture the multitude of concentration profiles available in the literature. The authors can deal with this by showing that their parametrization can be fitted to all temperature profiles of scenarios available in the IPCC scenario database (https://tntcat.iiasa.ac.at/AR5DB/), and the more recent SSP database (https://tntcat.iiasa.ac.at/SspDb/).

We thank the reviewer for this comment. We have undertaken the analysis as suggested, and this has created a new Supplementary Information for our paper. We have fitted our trajectory structure to all temperature profiles presented (by group and scenario) in the AR5 database, and also for the marker scenarios in the shared socioeconomic pathways database. This is subject to a criterion that the decadal temperature estimates for any particular projection show evidence of stabilisation. We set this as temperature remaining below three degrees above pre-industrial, and that the absolute difference in temperature between year 2090 and year 2100 is less than 0.1°C.

These additional calculations generate a new Supplementary Information Table S1,

repeated in full at the bottom of this response. In addition to presenting the fitted parameters, we also calculate the root-mean square deviations of the differences between model fit and our analytical. In all instances, we find this to be very small (order $0.02°C$) suggesting the curve structure is sufficiently versatile to fit model responses to multiple concentration profiles.

We now write in the manuscript a new paragraph under Section 2.3, as: "We additionally fit our curves to pathways in which emissions are generated using integrated assessment models (IAM), and related global temperature profiles created using a simple climate model. This is for warming profiles from the IPCC scenario database (https://tntcat.iiasa.ac.at/AR5DB/) and for the marker scenarios of the more recent shared socioeconomic pathways (SSP) database (https://tntcat.iiasa.ac.at/SspDb). We demonstrate that the functional forms used here can also be fitted to these IAM-based scenarios to a good level of accuracy (see Supplementary Information)."

2) A smaller point is the dependence of the framework on stabilizing temperatures. For impact studies, it would also be interesting to be able to explore pathways which gradually decline temperatures.

The mathematical form of the equations has been designed such that they will eventually tend towards convergence i.e. stabilisation. However it is possible that the time at which stabilisation is approached is beyond the horizon of any particular impacts study. Hence the curves can enable temperature to be either raising or falling through more immediate times of interest. The yellow curves Figure 1 for instance, are slow to stabilise. Based on this comment, we now write in the conclusions: "Where an impacts study is for a period ahead that is much less than the time to stabilisation, then these curves allow for the possibility of gradually rising or declining temperatures through any analysis period"

3) P1L3: The Paris Agreement is committed to holding the rise in global average temperature increase to "well below" 2âŮęC

We have added the word "well", as suggested.

4) P1L4-5: The "given emissions cuts to achieve the lower target may be especially difficult to achieve" argument is weak, and is not supported in the remainder of the manuscript. Depending on timing, similar emissions cuts are to be considered for 1.5âŮęC and 2âŮęC scenarios (but with a delay of 1 decade or so). In addition to the questions highlighted here, there is at least one more very important question, which is related to the reversibility of warming after an overshoot. I think this should also be mentioned.

Based on this remark, we have taken out these words (i.e. "given emissions. . ...."). We accept this could inadvertently be regarded as a judgement statement. This sentence now reads simply: "Second, what is the benefit from reduced climate impacts by keeping warming at or below 1.5°C?" Hence this now makes no statement on feasibility.

Regarding reversibility of warming, our profiles do allow for this (e.g. yellow curves in Figure 1, 2). Based on this comment, we re-iterate this more strongly in the Conclusions that these curves can allow overshoot. Again we do not present any view on feasibility of particular profiles, including ability to "get back" to lower temperatures. Hence we have adjusted the Conclusions to say: ". . ..through to stabilised temperature levels. They can include an initial overshoot of temperatures above any desired final warming level".

5) P1L7: The basis for this "implication" is weak. Until now, models have been run in forward mode and have been able to provide lots of useful information for limiting warming to and impacts at specific temperature limits.

We agree, and have adjusted this statement, acknowledging that there are studies that run in forward mode and that do provide useful information for impacts at different warming levels. We now simply say: "It is useful to operate models in invertible form, to make model-specific estimates of greenhouse gas (GHG) concentration pathways consistent with prescribed temperature profiles"

6) P1L14: Please specify that these are projections by different ESMs.

Done. We now write: "The curves capture temperature profiles from the existing rcp2.6 scenario projections by a range of different earth system models (ESMs), which. . .."

7) P2L3-4: It would be good to provide a reference for this claim.

This sentence has been re-written, and now includes two additional references. We now write: "..below a 1.5 degrees warming threshold. To achieve the latter could in particular involve major changes of energy demand or production (Rogelj et al., 2013), and extensive reliance on artificial carbon removal (Fuss et al., 2014) such as biofuels combined with carbon capture and storage."

8) P2L4-5: Not clear what the relevance is of this statement. The authors refer here to the purely academic case of constant concentrations. Such a case is arguably in practice even harder to achieve than eliminating emissions.

We respectfully request we retain this, as the idea of a constant stabilised concentration commitment is long-established. It was mentioned, in particular, in AR4-WG1.

9) P2L9-10: Inverse modelling also cannot answer these questions, because there is no way to ensure that pathways are supported by technologies.

We agree, and based on this comment, we state that our illustrative curves are primarily to aid discussion of different potential pathways. We do not attach any assessment of feasibility to any of them at this stage. To ensure there is no misunderstanding, we have amended the document. We feel this is possibly best in the discussion, and write: "At this stage, we do not associate any particular parameter combinations (or ranges) with their feasibility of fulfilment by society"

10) P2L14-15: Recent publications provide an overview of various methods of exploring differences between warming levels of 1.5 and 2°C (James et al, Wiley Interdisciplinary Reviews: Climate Change, 2017). It would be useful to situate the approach proposed here in the context of these various methods.
James et al. (2017) is an important paper in the debate, and sorry we missed it. We now write in our manuscript: "In the comprehensive review of methods to identify regional differences associated with alternative global warming targets, James et al. (2017) note pattern-scaling as a key technique. The accuracy of this interpolation system has been recently reviewed in detail by Tebaldi and Arblaster (2014) and with enhancements proposed by Herger et al. (2015). In the other approaches of James et al. (2017), the central issue remains as how to interpret existing simulations, that even for identical forcings, project a range of different future final warming levels."

11) P6L15ff: This section is unclearly written. Please consider rewriting it providing a bit background to how the suggested activities could be implemented.

We have adjusted this manuscript both in response to requests above, and with a new paragraph. The "Applications" section now has three paragraphs. The first builds on the references above and discussion of scaling. The second is now tidier, describing how our curves may encourage calculation of any related emissions profiles to fulfil them. Then, based on this comment, we now provide a new additional paragraph that describes how application of these curves may allow direct research project inter-comparison. The revised section is repeated in full below:

"Our profiles enable a common framework for discussion of warming trajectories that stabilise to pre-defined temperature limits. Regional climate change corresponding to these global temperatures can be estimated from interpolation of ESM projections (e.g. by pattern-scaling, Huntingford and Cox, 2000). Such scaling techniques can be linked to impacts models (e.g., Huntingford et al., 2010). In the comprehensive review of methods to identify regional differences associated with alternative global warming targets, James et al. (2017) note pattern-scaling as a key technique. The accuracy of this interpolation system has been recently reviewed in detail by Tebaldi and Arblaster (2014) and with enhancements proposed by Herger et al. (2015). In the other approaches of James et al. (2017), the central issue remains as how to interpret existing simulations, that even for identical forcings, project a range of different future

warming levels.

Emissions profiles can be calculated to fulfil the ESM-dependent radiative forcings associated with any prescribed global temperature stabilisation profile. These can include different mixtures of individual greenhouse gas emissions, whilst accounting for any perturbed land-atmosphere and ocean-atmosphere gas exchanges. The sum of the radiation changes for altered individual atmospheric greenhouse gas combinations must equal the ESM-dependent radiative forcing. Although our analytical forms are generic and can be calculated for any prescribed final stabilised temperature $\Delta$TLim, the emphasis here is placed on the 1.5°C or 2.0°C targets. This is due to their strong current discussion in policy circles regarding "clean energy" (e.g. Obama, 2017).

To understand the significance between stabilizing global warming at either 1.5°C or 2.0°C is a complex and multi-dimensional problem. There are implications for regional climate changes, impacts and for "allowable" emissions and including the range of potential mixes between emitted greenhouse gases. These factors will also depend on the time evolution of global warming towards such warming thresholds. Each of these issues requires study, and ideally in a way that enables findings to be compared in a common framework. The application of these curves is to work towards such a framework, by offering a set of possible future warming pathways for utility in research initiatives, and that can be readily defined through a limited set of parameters."

12) P2&6: More recent papers have shown limitations of pattern scaling (e.g. Tebaldi and Arblaster, Clim. Ch., 2016). It would be good to also discuss these more recently identified limitations in the context of the proposed approach.

The main component of our paper is to present a set of analytical curves for global temperature. This can be linked to pattern scaling, and we welcome the opportunity to address the merits and issues with the latter. We think it best to do this via the literature, both the suggested paper and also a more recent one by Herger et al (2015). We now write in the manuscript: "The accuracy of this interpolation system has been recently

reviewed in detail by Tebaldi and Arblaster (2014) and with enhancements proposed by Herger et al (2015)."

Technical corrections:

1) P7L8-9: This sentences seems incomplete. More open to scrutiny and discussion compared to what?

This sentence was poorly worded. We have rewritten it, and enhanced the sentence that follows it, so they sit properly together. The manuscript now says "Their relative simplicity makes them transparent, and open to discussion. If common temperature scenarios are adopted by a range of studies (by selection of $\mu 0$, $\mu 1$ and $\Delta$TLim values), this may allow easier comparison of either the impacts of, or emission to achieve, 1.5oC or 2.0oC warming stabilisation."

Please also note the supplement to this comment:
http://www.earth-syst-dynam-discuss.net/esd-2017-17/esd-2017-17-AC2-supplement.pdf

---

## Author Comment (AC4) · 19 May 2017

Please find attached a single file with our full responses to reviewers, in formatted version and with SI attached.

Please also note the supplement to this comment: http://www.earth-syst-dynam-discuss.net/esd-2017-17/esd-2017-17-AC4-supplement.pdf

---

## Author Response (AR1)

**CEH Wallingford**
Wallingford, OXON, OX10 8BB, U.K.
Email: chg@ceh.ac.uk
Tel: +44 (0)7884437138        June 4th 2017

The Editors
Earth System Dynamics
Copernicus Publications
Bahnhofsallee 1e
37081 Gottingen, Germany

Dear Editors,

Please find resubmitted final electronic files for paper:

**"Flexible parameter-sparse global temperature time-profiles that stabilise at 1.5°C and 2.0°C".**

We are pleased that our responses to reviewers' requests are satisfactory and that our manuscript is now formally accepted.

Please don't hesitate to contact me if you have any further questions. Thank you for all of your support and helping during this paper submission process.

With kind regards,

Dr Chris Huntingford and on behalf of co-authors